# Evaluation of Afamin Level after Bariatric Surgery in Patient with Obesity

**DOI:** 10.3390/jcm12030848

**Published:** 2023-01-20

**Authors:** Hilmi Demircioglu, Ugur Dogan, Hamit Yasar Ellidag, Arif Aslaner, Osman Zekai Oner

**Affiliations:** 1Department of General Surgery, Antalya Education and Research Hospital, University of Health Sciences, 07100 Antalya, Turkey; 2Central Laboratories, Antalya Education and Research Hospital, University of Health Sciences, 07100 Antalya, Turkey

**Keywords:** afamin, bariatric surgery, metabolic syndrome, laparoscopic sleeve gastrectomy, body mass index

## Abstract

Background: The aim of this study is to evaluate afamin levels after weight loss in obese patients who underwent laparoscopic sleeve gastrectomy (LSG) and to investigate the relationship between them. In addition, after bariatric surgery, thyroid stimulating hormone (TSH), thyroxine (T4), low-density lipoprotein (LDL), very low-density protein (VLDL), total cholesterol, triglyceride (TG), high-density lipoprotein (HDL), insulin, and hemoglobin A1c (HgbA1c) levels were evaluated. Methods: Preoperative and postoperative 6th month venous blood samples were obtained from 43 patients included in this study. The preoperative and postoperative 6th month body mass index (BMI), TG, total cholesterol, VLDL, HDL, insulin, HgbA1c, TSH, T4, and afamin levels of the patients who underwent bariatric surgery with obesity were compared. Results: Serum afamin levels of patients decreased at 6 months postoperatively; however, it was not statistically significant. We observed a statistically significant decrease in patients’ BMI, HDL, VLDL, TG, total cholesterol, TSH, T4, HgbA1c, and insulin values (*p* < 0.05). There were significant increases in HDL and T4 values. The change in LDL value was statistically insignificant. Conclusions: Recent studies have shown that there may be a cause–effect relationship between afamin and obesity. In our study, we observed a decrease in serum afamin levels after weight loss following bariatric surgery. In addition, we think that afamin may be used as a potential marker of metabolic syndrome in the future and may lead to improvements in the medical treatment of obesity.

## 1. Introduction

Obesity is the most important and common endocrine and metabolic disease worldwide. Morbid obesity has become an epidemic problem in the last hundred years, and it is the second leading cause of death after smoking addiction among preventable diseases. There are many factors that regulate energy intake and consumption. Obesity, a disease in which complex genetic and environmental factors play a role in its etiology, is characterized by excessive energy intake and an increase in the amount of adipose tissue. Grades of obesity are commonly defined by body mass index (BMI = body weight/square meter of height). Patients are classified as underweight, normal weight, overweight, obese, morbidly obese, and super obese (Table 1) [1].

Another way to define obesity is by the measurement of body adipose tissue. The presence of adipose tissue more than 25% of the total body weight in men and more than 30% in women can be defined as obesity. A third way to define obesity is to determine the waist-to-hip ratio. Depending on the effects of obesity, many co-morbid diseases occur. Hypertension, dyslipidemia, cardiovascular diseases, degenerative joint diseases, type 2 diabetes, gastroesophageal reflux disease (GERD), cholelithiasis, hypoventilation syndrome, certain types of cancer, pseudotumor cerebri, migraine headache, and increased mortality risk are diseases caused by obesity [2].

The most effective treatment method of obesity today is bariatric surgery. Bariatric surgery provides long-term permanent weight loss, reduces appetite and food intake. By providing glycemic control directly and indirectly, it reduces the metabolic effects of obesity and provides weight control. Thus, it prevents many co-morbidities and increases survival. Compared to most treatment modalities today, it has been observed that maintenance of weight loss can often be achieved with bariatric surgery. Bariatric surgery can help obese patients to lose even more than 50% of their excess weight [3].

Afamin is a vitamin E-binding glycoprotein from the albumin gene family and it is expressed in the liver. It is a transport protein that carries hydrophobic molecules by acting in an albumin-like way. It is considered to play a role in antiapoptotic and antioxidant processes. It is mainly secreted from the liver; however, it can also be secreted from the brain, kidneys, testicles, and ovaries [4,5]. In vitro studies have revealed the specific binding patterns between vitamin E and afamin. Its abundant presence in the ovarian follicular and cerebrospinal fluid suggests that it plays a significant role in fertility and neuroprotection. Today, the role of afamin in the pathogenesis of metabolic syndrome and obesity development as well as its relationship with appetite hormones are being investigated in human and animal models. The anatomical shrinkage of the stomach and subsequent hormonal changes related to appetite and the interaction of brain-derived afamin levels may have an effect on obesity [6]. A study showed that afamin values increased in correlation with the increase in body mass index of patients with gestational diabetes mellitus [7], whereas another study indicated that serum afamin values were associated with all metabolic syndrome components [8].

The main purpose of our study was to evaluate afamin levels after weight loss in obese patients who underwent bariatric surgery and to investigate the relationship between them. In addition, after bariatric surgery in obese patients, thyroid stimulating hormone (TSH), thyroxine (T4), low-density lipoprotein (LDL), very low-density protein (VLDL), total cholesterol, triglyceride (TG), high-density lipoprotein (HDL), insulin, and hemoglobin A1c (HgbA1c) levels were evaluated. We also think that afamin can be used as a potential metabolic syndrome marker in the future and may lead to improvements in the medical treatment of obesity.

## 2. Materials and Methods

In this study, 43 obese patients with a BMI over 40 or over 35 with comorbidities to undergo bariatric surgery were included. The preoperative and postoperative 6th month Afamin, total cholesterol, TG, LDL, HDL, VLDL, fasting insulin, T4, TSH, and HgbA1c values of the patients were compared and the correlation between weight loss and changes in these parameters was examined. Approval was obtained from the Health Sciences University Antalya Training and Research Hospital Clinical Research Ethics Committee (22/10/2020–protocol number 16/5) and there was adherence to the tenets of the Declaration of Helsinki. Laparoscopic sleeve gastrectomy (LSG) procedure was applied to the patients in our General Surgery Clinic, which met the basic requirements of the “Center of Excellence” determined by the International Federation for the Surgery of Obesity and Metabolic Disorders (IFSO). A total of 43 patients, 32 female and 11 male, aged between 18–59 years, who were operated between October 2020 and March 2021 were included in the prospectively planned study. In our center with a high patient volume, the treatment of the patients was performed in a multidisciplinary manner. We interrogated the demographics, personal backgrounds, and family backgrounds of the patients by taking a detailed medical history, measuring thyroid function tests (TFT), cholesterol, fasting insulin, and HgbA1c values within the scope of our study, and performing abdominal ultrasonography and endoscopies. Among the clinical parameters, the patient’s age (year), weight (kg) and height (cm), body mass index, length of stay in hospital, postoperative hemogram and routine biochemistry values, and vital records were saved. Patients aged 18 years and over who underwent LSG with the diagnosis of obesity in our clinic were included in this study. Exclusion criteria were patients who did not accept to participate in this study, patients under the age of 18 and over the age of 59, those who had undergone previous abdominal surgery, or those who had undergone revision surgery due to complications.

### 2.1. Surgical Method

The patient was placed on the operating table in supine position, antiembolic stockings were put on, and a foley catheter was inserted. The surgeon was positioned between the patient’s legs, the cameraperson was on the right side, and the assistant surgeon was on the left side. A total of five trocars were positioned: a 10 mm trocar from the middle line 25 to 30 cm below the xiphoid and above the umbilicus, a 5 mm trocar from the left anterior axillary line, a 12 mm trocar from the left midclavicular line, a 10 mm trocar from the left midclavicular line, and a 5 mm trocar below the xiphoid process. After pneumoperitoneum had been achieved, the phrenoesophageal ligament was divided and the left crus of the diaphragm was identified. The angle and proximal stomach were separated from upper part of the spleen using a laparoscopic vessel sealing device (Covidien, Minneapolis, MN, USA). The short gastric vessels on the great curvature of the stomach were divided to allow entrance into the omental bursa. After this division had been advanced 4 cm proximal to the pylorus, a 32 French size bougie was passed through the mouth and placed along the small curvature of the stomach. Since the gastric wall thickness is anatomically different, the stomach was divided into two with the appropriate stapler and a gastric tube was created. The resected stomach was then removed through the 12 mm trocar hole. Finally, the entire staple line was washed, the remaining fluid was aspirated with saline, and a drainage tube was placed across the stomach. The trocar entrances were closed using fascia 2-0 vicryl to prevent herniation. Skin It was sutured using 3-0 prolene.

### 2.2. Measurement of Biochemical

As for the collection of blood samples, antecubital venous blood samples (approximately 10 mL) were obtained from all patients included in this study on the morning of the operation day. The blood samples collected from each participant were put into ethylenediamine tetraacetic (EDTA) tubes with added aprotinin and taken to the laboratory within 30 min. The blood samples taken for the measurement of afamin level were centrifuged in Nüve (Nüve Sanayi Malzemeleri İmalat ve Ticaret A.Ş, Ankara, Turkey) NF800 centrifuge devices at 4000 rpm for 10 min, and following the separation of the serum part, the serum samples were transferred to Eppendorf tubes and stored at −80 °C until the study day. Insulin (fasting), HgbA1c, TSH, T4, total cholesterol, triglyceride, LDL, HDL, and VLDL serum values within the scope of this study were taken during the preoperative evaluation one week before the surgery. Antecubital venous samples (approximately 25 mL) were collected at 6 months postoperatively while the patients had an empty stomach. The serum part of the blood samples collected for the measurement of the afamin level was separated and stored at −80 °C. Insulin (fasting), HgbA1c, TSH, T4, total cholesterol, triglyceride, LDL, HDL, and VLDL values were evaluated routinely in the hospital laboratory. Afamin levels were measured by the enzyme-linked immunosorbent assay (ELISA) method. Afamin was stored at room temperature for about half an hour with Human Afamin ELISA kits manufactured by AFG Scientific (catalog number EK 714650, AFG Bioscience LLC, USA). After washing with an Awareness brand Stat Fax 2600 device, afamin levels were evaluated with an Awareness brand ChroMate 4300 device and immunoassay 450 wavelength method. The values were measured according to the manufacturer’s instructions.

### 2.3. Statistical Analysis

Descriptive statistics are presented with frequency, percentage, mean, standard deviation, median, minimum, and maximum values. The assumption of normality was checked with the Shapiro–Wilk test. When analyzing the difference between the numerical data measured preoperatively and postoperatively, the paired *t*-test was used in cases where the data were in a normal distribution or the Wilcoxon signed-rank test in opposite cases. Relationships between numerical data were evaluated by the non-parametric Spearman correlation test. Analyses were performed with the Statistical Package for Social Sciences (SPSS) version 23.0 (IBM Corp., Armonk, NY, USA) program. A value of *p* < 0.05 was considered statistically significant. ROC analyses were used to determine the cut-off value for the afamin level.

## 3. Results

We analyzed the statistical data of our patients who underwent sleeve gastrectomy. We also analyzed the co-morbidities, age, height and BMI values, preoperative and postoperative 6th month afamin, total cholesterol, HDL, LDL, VLDL, triglyceride, insulin, HgbA1c, TSH, and T4 values of our patients participating in this study.

The mean age of the 43 patients included in this study was 36.97 ± 11.21, mean height measurements were 163.20 ± 7.23 cm. Table 2 shows the sex distribution of the patients participating in this study, whether they have any co-morbidities, and, if yes, the name of the co-morbidities. According to this table, 74.4% of the 43 patients participating in this study were female and 25.6% were male. Co-morbidities were observed in 46.56% of the patients while 53.5% of them did not have any. The most common diseases were diabetes mellitus (DM) with 27.9%, hypertension (HT) with 18.6%, and hypothyroid with 13.9%.

The preoperative and postoperative biochemical parameters of obese patients who underwent bariatric surgery are shown in Table 3.

Although there was a decrease in afamin levels at 6 months postoperatively, this decrease was not statistically significant. (*p* = 0.17) (Figure 1). There was a statistically significant difference between preoperative insulin, HgbA1c, TSH, T4, total cholesterol, TG, HDL, and VLDL values and postoperative 6th month values (*p* < 0.05). A statistically significant decrease was observed in insulin values, HgbA1c, TSH, total cholesterol, TG, and VLDL values of postoperative patients. On the other hand, there was a statistically significant increase in T4 and HDL values. There was no statistical difference between preoperative LDL values and postoperative LDL values (*p* = 0.056).

There was a statistically significant difference between the preoperative body weights of the patients and their postoperative 6th month body weights (*p* < 0.0001). Similarly, the comparison of BMI values also showed a statistical difference. The body weights of the patients and the BMI values calculated accordingly decreased in the postoperative 6th month (Table 4).

There was no statistically significant relationship between the differences in preoperative and postoperative body mass index (BMI), the difference in body weight, and the difference in postoperative and preoperative afamin values (*p* = 0.733 and *p* = 0.568, respectively) (Table 5).

The comparison of the difference in BMI, body weight, and afamin values in the preoperative and postoperative 6th month according to sex and the presence of co-morbidities is given in Table 6. No statistically significant difference was observed in terms of the difference in BMI, body weight, and afamin values according to sex and the presence of co-morbidities.

## 4. Discussion

Obesity is a serious disease with a high incidence of mortality and morbidity worldwide. Obesity has an insidious, sudden and mortal course with serious complications, making it one of the diseases that must be treated definitively. Genetic, psychological, endocrine, and cultural factors play a role in the etiology of the disease [9]. Although most obese patients have some complaints, they can continue their lives close to normal. Risk factors for each of the complications that develop with obesity and metabolic syndrome overlap, and the risk of complications increases as the degree of obesity increases. The damage caused by the most common of these complications, such as DM and HT, to the vascular system is usually detected after the complications have developed; however, endothelial dysfunction predisposing to myocardial infarction (MI) can start from the age of two. The results of the complications that may occur in the individual with the genetic predisposition, insulin resistance, and the pathophysiological processes that advance by making an addictive effect on each other are obesity, metabolic syndrome, and many diseases caused by them. An excessive diet rich in fat and carbohydrates and a sedentary lifestyle can cause metabolic syndrome, leading to systemic inflammation, insulin resistance, and the development of type 2 diabetes. It is likely that there are factors in the body that can detect the metabolic syndrome in the blood and be used as predictive biomarkers of the disease. Changes in serum concentration levels of the biomarkers to be used should be detectable in the early stages of potential complications and should correlate with the progression of the disease and obesity. Identification of such biomarkers enables early intervention in the disease and helps to minimize the potential negative consequences by monitoring the identified biomarkers to return to normal values after the regression of obesity. In addition to the biomarkers, certain newly identified molecules may also play a role in the development of the disease. There are studies investigating the effects of bariatric surgery on proteins that may be associated with obesity, thyroid hormones closely related to metabolism, and blood lipid profile [10,11,12]. Unfortunately, although the biomarkers used in the prediction and diagnosis of morbid obesity to date contribute to the prognosis and management of many complications of the disease, an effective biomarker in the direct control of obesity is still unknown. In this study, afamin levels were evaluated after weight loss in obese patients who underwent laparoscopic sleeve gastrectomy and the relationship between them was investigated. In addition, after bariatric surgery, thyroid stimulating hormone (TSH), thyroxine (T4), low-density lipoprotein (LDL), very low-density protein (VLDL), total cholesterol, triglyceride (TG), high-density lipoprotein (HDL), insulin, and hemoglobin A1c (HgbA1c) levels were evaluated.

It has been shown that the organ in which afamin, a glycoprotein discovered as the fourth molecule in the albumin gene family, is primarily synthesized and released into circulation is the liver. Although much of the physiology and function of afamin still remains a mystery, its affinity for vitamin E and its similarity to albumin suggests that it has a function to transport hydrophobic molecules [13]. A study showed that afamin carries vitamin E across the blood–brain barrier in vitro cell culture and also protects neurons against oxidative stress with its antioxidant function [5]. In another study, the in vitro effects of afamin and vitamin E on neuronal cells were investigated. As a result of the study, it was observed that afamin and vitamin E synergistically increased the survival of cortical neurons under apoptotic conditions and showed a neuroprotective activity [14].

Another study indicated that afamin doubled during pregnancy and decreased to normal values after delivery. It was thought that this increase was due to the increase in the hormones controlling the secretion of afamin during pregnancy. Although it was primarily thought that afamin was secreted against oxidative stress, the presence of afamin in the placental tissue could not be shown. In this case, it was thought that afamin was secreted against maternal stress [15]. A study conducted with Holstein cows showed that the level of afamin decreased when colostrum was secreted during lactation. The reason for this decrease was thought to be the transfer of high levels of vitamin E from the colostrum to the calf. The time when the afamin level is lowest coincides with the time of full colostrum secretion. In addition, the study did not detect mRNA transcription of the afamin gene in the mammary gland tissue [16].

Afamin has been found to be strongly associated with all components of metabolic syndrome [8,17]. Transgenic mice overexpressing the human afamin gene showed increased body weight and increased lipid and glucose concentrations. These transgenic mice data yielded the same results as human population-based studies showing that afamin was strongly associated with the prevalence and development of metabolic syndrome [8,18]. In the aforementioned study, plasma concentrations of afamin were positively correlated with waist circumference, BMI, triglyceride, systolic and diastolic blood pressures, LDL, total cholesterol, and blood glucose while they showed a weak negative correlation with HDL, whereas in our study, HDL values increased significantly while triglyceride, LDL, and total cholesterol values in correlation with the decrease in BMI regressed. Although there was a decrease in the level of afamin, it was not statistically significant. Interestingly, in animal studies, the offspring of the transgenic mice in the control group did not show metabolic syndrome symptoms despite high afamin levels during adolescence. The fact that afamin does not cause metabolic syndrome despite its high concentration in adolescence suggests that it has a growth hormone-like effect on metabolism [8]. Again, another study found a correlation between type 2 diabetes and afamin values, and it was thought that afamin could be a new marker for the follow-up of individuals at risk of type 2 diabetes [19]. The findings of the study showed that the high concentrations of oxidative stress in individuals with a high BMI were associated with metabolic syndrome, insulin resistance, and type 2 diabetes [17]. In our study, we observed that the components forming the metabolic syndrome regressed with bariatric surgery and there was a significant decrease in HgbA1c values of both type 2 diabetes patients and prediabetics. In a study conducted with 30 patients with metabolic syndrome in Iran, nano-curcumin therapy was administered for 2 months. While the results of the study showed no significant decrease in terms of afamin and BMI, there was a significant improvement in blood pressure values. It was observed that HDL and afamin values were inversely correlated; however, there was no significant decrease in afamin [20]. Similar to our study, it was observed that there was a decrease in afamin level in patients with weight loss and decreased BMI, but this decrease was not statistically significant. These results have raised the question whether high levels of afamin may be genetic just as there is a genetic predisposition of obese people with metabolic syndrome.

In a study conducted with patients diagnosed with preeclampsia or only hypertension during pregnancy, afamin levels were found to be twice as high in pregnant women with hypertension. Moreover, the pre-pregnancy BMI values of these patients were within the normal range. It has been suggested that afamin levels that are already elevated during pregnancy may cause hypertension by mechanisms independent of obesity just as beta-hCG causes in certain studies [15]. While certain studies showed a significant correlation between afamin and polycystic ovary syndrome (PCOS), no significant correlation was observed in many others [21]. A study conducted by Dieplinger H. et al. showed that while the afamin values were significantly higher in women who had PCOS and insulin resistance than in women who had PCOS but no insulin resistance, they were not significantly higher in women who had PCOS alone compared to the control group. In the univariate analyses in their study, serum concentrations of afamin were significantly correlated with BMI, triglycerides, homeostatic model assessment (HOMA) index, and fasting insulin [22]. In multivariate analyses, only triglyceride concentration was found to be an independent predictor of afamin. In the previous study, subjects with metabolic syndrome were evaluated as having higher median concentrations of afamin than those without metabolic syndrome (77.43 ± 28,60 mg/L and 65.08 ± 18.03 mg/L, *p* = 0.010) [18]. In our study, insulin, BMI, and triglyceride levels were significantly decreased. However, the decrease in afamin levels was not statistically significant.

In another study, the relationship between serum afamine and LDL, total cholesterol and vitamin E levels was investigated [19]. After LDL apheresis, there was a 61% reduction in LDL cholesterol and a 50% reduction in total cholesterol, with only 10% reduction in afamin level. A decrease of 14% in HDL cholesterol and 30% in vitamin E levels was observed. That study predicted that LDL apheresis may be effective in reducing cardiac risk with its effect on dyslipidemia. In our study, while the blood lipid profile of the patients improved after weight loss, a statistically insignificant decrease was observed in afamin. While HDL decreased in the previous study, HDL increased significantly in our study. In our study as well as in many others, HDL value change was found to be associated with weight loss and diet change [23]. LDL and total cholesterol, the height of which is considered as dyslipidemia, decreased statistically in both our study and the previous study, while the decrease in afamin was not significant in either study.

A study showed that the AFM gene was also expressed in the kidneys. In a patient with nephropathy, it was found that the urinary afamin level and afamin creatinine ratio increased significantly. The authors expressed their opinion that afamin could be used as a biomarker for kidney damage [24]. A study conducted with a similar but larger population indicated a significant increase in urinary afamin in primary membranous nephropathy [25]. Studies conducted with urinary afamin in chronic kidney diseases found that afamin increased significantly [26]. Another study showed that afamin was also significantly higher in the peritoneal fluid of patients with endometriosis. This increase raises the question of whether afamin increases as a protective protein in any part of the body with inflammation [27].

There are studies showing that afamin may have antioxidant and neuroprotective functions, and it was previously found to be involved in neuroprotection [14]. Moreover, a study conducted on patients with idiopathic nephrotic syndrome in the pediatric population found that urinary afamin was significantly high. Those who conducted the study thought that it was related to the high levels of oxidative stress [28]. The development of insulin resistance in obese people and the inflammation process that begins with obesity may be the cause of elevated serum afamin. On the other hand, it is also discussed whether high afamin levels cause systemic inflammation. Large epidemiological studies are needed to fully elucidate the role of afamin in pathologies associated with metabolic syndrome such as obesity, diabetes mellitus, and hepatic steatosis [20].

There are studies showing that afamin contains significant changes that can be used as a biomarker in both diagnosis and prognosis in certain types of cancer. Some of these cancer types have also been found to be associated with obesity. Serum afamin values have also been investigated in various types of carcinomas, including stomach, colorectal, cervix, liver, biliary tract, breast, and thyroid cancer [29,30,31]. There are studies showing that afamin can be used as a marker in ovarian malignancies whose relationship with obesity has been investigated many times. In addition, these studies found a correlation between higher afamin concentrations and survival time [32]. Further studies are needed for other types of cancer. The cause-and-effect relationship between obesity, afamin level, and cancer is not yet known in full details. In many studies, it has been shown that afamin level increases in direct proportion to BMI. Existing questions of whether the interaction is between BMI and cancer or a direct relationship between afamin level and cancer will be clarified as a result of new studies to be conducted in the future.

In our study, BMI, fasting insulin, HgbA1c, TSH, total cholesterol, triglyceride, and VLDL values decreased significantly after bariatric surgery. The decrease in LDL levels was not statistically significant. T4 and HDL levels increased significantly. Although afamin value decreased, it was not found to be significant. There are many studies showing that afamin is present at high levels by showing a positive correlation with the components of obesity and metabolic syndrome. We think that diet may be one of the reasons why the degree of weight loss and the components of metabolic syndrome accompanying obesity did not differ to the same extent in our study. In another study, it was found that the level of tocopheral transfer protein (TTPA), the protein carrying vitamin E, varies with diet. In another study, it was shown that TTPA gene and afamin gene expressions in the gastrointestinal tract are related to the level of vitamin E taken. This situation also suggested that in our study, afamin values in patients whose BMI decreased following bariatric surgery may have been affected by genetics or vitamin-fortified diets. We think that there is a need for long-term follow-up studies with larger numbers of patients to reveal the metabolic effects of afamin in the human body and its relationship with obesity.

## 5. Conclusions

In this study, afamin levels were evaluated after weight loss in obese patients who underwent laparoscopic sleeve gastrectomy and the relationship between them was investigated. In addition, after bariatric surgery, thyroid stimulating hormone (TSH), thyroxine (T4), low-density lipoprotein (LDL), very low-density protein (VLDL), total cholesterol, triglyceride (TG), high-density lipoprotein (HDL), insulin, and hemoglobin A1c (HgbA1c) levels were evaluated.

When the postoperative 6th month afamin values of the patients were compared with the preoperative values, a decrease was observed but it was not found to be statistically significant. We think that the increase in sample size and follow-up period may affect the decrease in afamin levels. In our study, a statistically significant decrease was found when the preoperative and postoperative 6th month weights and BMI of the patients were compared. Significant improvement was observed in the lipid profile of the majority of patients with preoperative dyslipidemia, one of the parameters we evaluated within the scope of our study. While there was a statistically significant decrease in total cholesterol, VLDL, and TG values, the reduction in LDL cholesterol values was not significant. HDL cholesterol value, on the other hand, showed a statistically significant increase. A significant decrease was found in HgbA1c values in some of the patients with diabetes mellitus as a co-morbid disease. Moreover, a significant decrease was observed in fasting insulin values. When we evaluated the thyroid function tests of the patients, we found that TSH values decreased significantly and T4 values increased significantly.

Many clinical studies have shown that bariatric surgery can treat and control obesity and metabolic syndrome more effectively and for a longer time than other treatment methods. It has been understood that diabetes can be prevented from developing with effective weight loss in patients. New clinical studies investigating the relationship between afamin and energy balance will make a great contribution in the fight against obesity. We think that there is a need for comprehensive new studies involving more patients with long-term follow-up in order to use afamin as a prognostic factor and an important marker in the follow-up and treatment of obesity in the future.

## Figures and Tables

**Figure 1 jcm-12-00848-f001:**
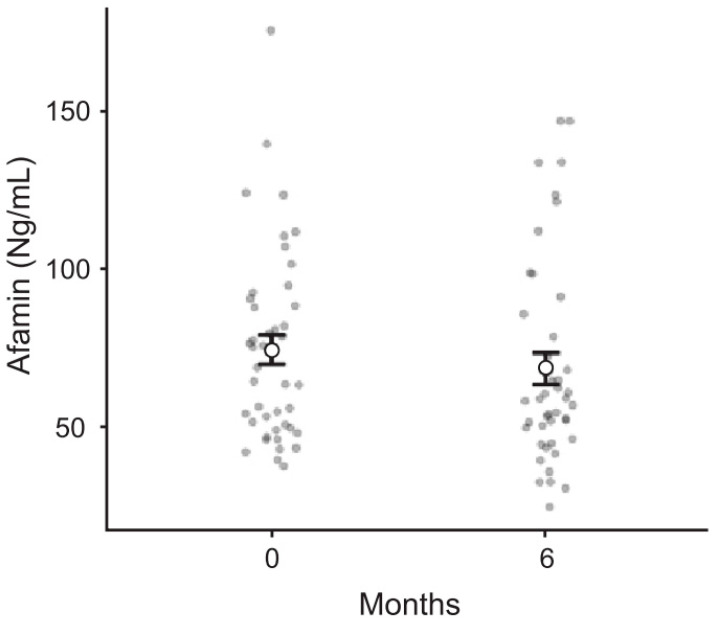
Preoperative and postoperative 6th month plasma afamin levels of obese patients.

**Table 1 jcm-12-00848-t001:** Definition of underweight, normal, overweight, and obese based on body mass index values.

Weight Category	Body Mass Index (kg/m²)
Underweight	<18.5
Normal	18.5–24.9
Overweight	25–29.9
Obese	>30
1st Degree 2nd Degree 3rd Degree (morbid)	30–34.9 35–39.9 >40

**Table 2 jcm-12-00848-t002:** Demographic and clinical characteristics of the patients with obesity.

		Number (*n*)	Percentage (%)
Sex	Female	32	74.4
	Male	11	25.6
Do you have any co-morbidities?	None	23	53.50
	Yes	20	46.56
Co-morbidity-DM	None	31	72.1
	Yes	12	27.9
Co-morbidity-HT	None	35	81.4
	Yes	8	18.6
Co-morbidity-Depression	None	42	97.7
	Yes	1	2.3
Co-morbidity-Asthma	None	39	90.7
	Yes	4	9.3
Co-morbidity-Hypothyroid	None	37	86
	Yes	6	13.9
Co-morbidity-CAD	None	42	97.7
	Yes	1	2.3
Co-morbidity-Hyperlipidemia	None	42	97.7
	Yes	1	2.3
Co-morbidity-Pituitary adenoma	None	42	97.7
	Yes	1	2.3
Co-morbidity-Bipolar	None	42	97.7
	Yes	1	2.3

Abbreviations: DM, diabetes mellitus; HT, hypertension; CAD, coronary artery disease. Multiple options have been marked for co-morbidities.

**Table 3 jcm-12-00848-t003:** Preoperative and postoperative 6th month comparison of blood parameters.

	Mean	SS	Median	IQR	Min.	Max.	*p* Value
Afamin (ng/mL) (preoperative)	74.4	30.14	68.9	40.98	37.83	175.33	0.17 ^b^
Afamin (ng/mL) (postoperative)	66.65	30.33	58.18	32.18	24.9	146.73	
Insulin (IU) (preoperative)	14.66	7.27	15.34	10.79	3.93	34.26	<0.0001 ^a^
Insulin (IU) (postoperative)	5.89	2.81	5.33	3.65	1.77	12.1	
HgbA1c% (preoperative)	6.38	1.03	6.1	0.75	5.1	9.1	<0.0001 ^b^
HgbA1c% (postoperative)	5.63	0.67	5.4	0.5	4.9	8.4	
TSH (uIU/mL) (preoperative)	3.55	5.05	2.3	1.73	0.9	31.81	<0.0001 ^b^
TSH (uIU/mL) (postoperative)	2.00	1.33	1.8	1.34	0.47	7.8	
T4 (ng/dL) (preoperative)	0.81	0.13	0.81	0.22	0.54	1.01	<0.0001 ^a^
T4 (ng/dL) (postoperative)	0.92	0.13	0.91	0.16	0.58	1.2	
Total Cholesterol (mg/dL) (preoperative)	196.25	44.13	195	69	99	280	0.049 ^a^
Total Cholesterol (mg/dL) (postoperative)	184.10	46.23	183	57	72	301	
TG (mg/dL) (preoperative)	141.55	71.24	122	67.5	31	395	0.003 ^b^
TG (mg/dL) (postoperative)	114.88	45.69	101	64.5	58	218	
HDL (mg/dL) (preoperative)	45.53	10.24	44.5	15.5	29	72	0.001 ^b^
HDL (mg/dL) (postoperative)	51.78	11.30	50.5	16	34	79	
LDL (mg/dL) (preoperative)	121.13	38.50	118.5	61.5	41	187	0.056 ^a^
LDL (mg/dL) (postoperative)	111.98	35.17	107	36.5	46	208	
VLDL (mg/dL) (preoperative)	29.83	14.47	26.5	19	11	79	0.001 ^b^
VLDL (mg/dL) (postoperative)	22.95	10.01	21	11.5	10	48	

^a^: Paired *t*-test, ^b^: Wilcoxon signed- rank test was used. Abbreviations: HgbA1c, hemoglobin A1c; TSH, thyroid stimulating hormone; T4, thyroxine; TG, triglyceride; HDL, high-density lipoprotein; LDL, low-density lipoprotein; VLDL, very low-density protein.

**Table 4 jcm-12-00848-t004:** Comparison of preoperative and postoperative 6th month body weight and BMI values.

	Mean	SS	*p*
Body weight (kg) (preoperative)	117.06	13.18	<0.0001
Body weight (kg) (postoperative)	95.87	11.16	
BMI (kg/m^2^) (preoperative)	43.93	5.08	<0.0001
BMI (kg/m^2^) (postoperative)	36	4.6	

Abbreviations: BMI, body mass index.

**Table 5 jcm-12-00848-t005:** The relationship between weight loss, BMI, and afamin level.

		Difference in Afamin Value (Preoperative and Postoperative)
Difference in BMI (preoperative and postoperative) (kg/m^2^)	r	0.053
	*p*	0.733
Difference in body weight (preoperative and postoperative) (kg)	r	0.090
	*p*	0.568

r: Correlation coefficient; Abbreviations: BMI, body mass index.

**Table 6 jcm-12-00848-t006:** Comparison of differences in BMI, body weight, and afamin values according to sex and presence of co-morbidities.

	Sex	*n*	Mean	SS	Minimum	Maximum	*p* Value
Difference in BMI (preoperative and postoperative) (kg/m^2^)	Female	32	7.52	2.83	3.01	13.85	0.356
	Male	11	9.13	4.57	4.85	18.11	
Difference in body weight (preoperative and postoperative) (kg)	Female	32	19.61	6.75	8.7	32.2	0.241
	Male	11	25.78	14.81	13.6	60	
Difference in afamin (preoperative and postoperative) (ng/mL)	Female	32	1.23	80.17	−120.73	370.98	0.466
	Male	11	3.5	31.46	−35.47	69.85	
	Do you have any co-morbidities?						
Difference in BMI (Preoperative and postoperative) (kg/m^2^)	None	23	15.77	88.87	−120.73	370.98	0.480
	Yes	20	−14.25	36.83	−81.4	58.3	
Difference in body weight (preoperative and postoperative) (kg)	None	23	21.29	11.71	9.5	60	0.495
	Yes	20	21.08	6.88	8.7	35.6	
Difference in afamin (preoperative and postoperative) (ng/mL)	None	23	15.77	88.87	−120.73	370.98	0.154
	Yes	20	−14.25	36.83	−81.4	58.3	

The Mann–Whitney U test was used. Abbreviations: BMI, body mass index.

## Data Availability

The data used and analyzed during this research are available from the corresponding author upon reasonable request.

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
