# Peer review of "Evaluation of Afamin Level after Bariatric Surgery in Patient with Obesity"

_jcm, 2023, doi:10.3390/jcm12030848_

Round 1

Reviewer 1 Report

This is an interesting and well written study. The science is fine but I am concerned about the conclusions. They state that all parameters except afamin decrease but yet their conclusions are based on a drop in afamin. There was no statistically significant drop in afamin. It is hardly surprising that the other parameters all dropped. Why base the conclusions on the one parameter which did not drop?

Author Response

Manuscript Number: jcm-2142454

Title: Evaluation of Afamin Level after Bariatric Surgery in Patient with Obesity 
Journal: Journal of Clinical Medicine
Authors: Hilmi Demircioglu, Ugur Dogan, Hamit Yasar Ellidag, Arif Aslaner, Osman Zekai Oner 

To the Editor, 
We thank you for your reviews. Our study will have its final and best form according to your reviews. We are grateful for your contributions. 
The authors appreciate the work done on our paper. 
Below, please find point-by-point replies to the concerns raised by the reviewers.
In addition, according to the reviewers’ comments we revised the manuscript and indicated the necessary changes as red in the paper. We hope that you would consider our manuscript for publication in the Journal of Clinical Medicine.
We confirm that all author details on the revised version are correct, that all authors have agreed to authorship and order of authorship for this manuscript and that all authors have the appropriate permissions and rights to the reported data.
Best Regards,

Corresponding author: Ugur Dogan
Department of General Surgery, Antalya Education and Research Hospital University of Health Sciences, Varlık Mahallesi, Kazim Karabekir Caddesi, 07100 Antalya/TURKEY
Phone: +90 532 6335686
e-mail: [email protected]

Reviewer comments:
Reviewer #1: 
We are grateful for your contributions. 
This is an interesting and well written study. The science is fine but I am concerned about the conclusions. They state that all parameters except afamin decrease but yet their conclusions are based on a drop in afamin. There was no statistically significant drop in afamin. It is hardly surprising that the other parameters all dropped. Why base the conclusions on the one parameter which did not drop?
Revision required was made in the text (Abstract, introduction and discussion). 
Paragraph rearranged:
''The main purpose of our study is to evaluate afamin levels after weight loss in obese patients who have undergone bariatric surgery and to investigate the relationship between them. In addition, after bariatric surgery in obese patients, thyroid stimulating hormone (TSH), thyroxine (T4), low-density lipoprotein (LDL), very low-density protein (VLDL), total cholesterol, triglyceride (TG), high-density lipoprotein (HDL) , insulin and hemoglobin A1c (HgbA1c) levels were evaluated.''

Reviewer 2 Report

Dear editor, 

I would to thank you to let me read this interesting paper about relationship between afamin and post-operative results after bariatric surgery.

This paper is interesintg bu there are several issues which need to be corrected before publication in JCM

First of all, title is not appropriate because changes in afamine level are not statistically significant, title might be changed to be more specific.

Second, introduction is correct no issues in it

Third, method section, there are some lacking informations :

period of the study, type of the bariatric center, volume of patients who underwent PBS in this center, sex ratio, type of study (retrospective), if patient underwent surgery consecutively

fourth : statistical method : with 43 patients you can't perform many tests as your did

finally, main issue is the number of patients, with only 43 patients, beta risk is high and afamin test is not significant. you should include more patients if you can and maybe your result will change

Yours sincerely

Author Response

Manuscript Number: jcm-2142454

Title: Evaluation of Afamin Level after Bariatric Surgery in Patient with Obesity 
Journal: Journal of Clinical Medicine
Authors: Hilmi Demircioglu, Ugur Dogan, Hamit Yasar Ellidag, Arif Aslaner, Osman Zekai Oner 

To the Editor, 
We thank you for your reviews. Our study will have its final and best form according to your reviews. We are grateful for your contributions. 
The authors appreciate the work done on our paper. 
Below, please find point-by-point replies to the concerns raised by the reviewers.
In addition, according to the reviewers’ comments we revised the manuscript and indicated the necessary changes as red in the paper. We hope that you would consider our manuscript for publication in the Journal of Clinical Medicine.
We confirm that all author details on the revised version are correct, that all authors have agreed to authorship and order of authorship for this manuscript and that all authors have the appropriate permissions and rights to the reported data.
Best Regards,

Corresponding author: Ugur Dogan
Department of General Surgery, Antalya Education and Research Hospital University of Health Sciences, Varlık Mahallesi, Kazim Karabekir Caddesi, 07100 Antalya/TURKEY
Phone: +90 532 6335686
e-mail: [email protected]

Reviewer comments:
Reviewer #2: 
We are grateful for your contributions. 
I would to thank you to let me read this interesting paper about relationship between afamin and post-operative results after bariatric surgery.
This paper is interesintg bu there are several issues which need to be corrected before publication in JCM.

First of all, title is not appropriate because changes in afamine level are not statistically significant, title might be changed to be more specific.
The title of our study was changed as "Evaluation of Afamin Level after Bariatric Surgery in a Patient with Obesity" in line with your suggestions.

Second, introduction is correct no issues in it
Third, method section, there are some lacking informations :
period of the study, type of the bariatric center, volume of patients who underwent PBS in this center, sex ratio, type of study (retrospective), if patient underwent surgery consecutively
Revision required was made in the text (Method section). 
Paragraph rearranged:
''Laparoscopic sleeve gastrectomy (LSG) procedure was applied to the patients in our General Surgery Clinic, which met the basic requirements of the "Center of Excellence" determined by the International Federation for the Surgery of Obesity and Metabolic Disorders (IFSO).  A total of 43 patients, 32 female and 11 male, aged between 18-59 years, who were operated between October 2020 and March 2021, were included in the prospectively planned study. In our center with a high patient volume, the treatment of the patients was performed multidisciplinary.'' 
''Patients aged 18 years and over who underwent LSG with the diagnosis of obesity in our clinic were included in the study. Exclusion criteria were patients who did not accept to participate in the study, patients under the age of 18 and over the age of 59, who had undergone previous abdominal surgery, or who had undergone revision surgery due to complications.'' 

fourth : statistical method : with 43 patients you can't perform many tests as your did
finally, main issue is the number of patients, with only 43 patients, beta risk is high and afamin test is not significant. you should include more patients if you can and maybe your result will change

Revision required was made in the text.
Thank you for your comment that the number of our patients is limited and the results may change when it is performed with more patients. However, as we carried out our study during the Covid-19 pandemic, when elective surgical operations and routine patient follow-up were very difficult, the number of our patients was 43. We think that new studies with a larger number of patients will contribute to the results.

Round 2

Reviewer 1 Report

The revised manuscript is now much better. The concerns have been addressed.

Reviewer 2 Report

All of my queries were done. So i recommend to publish this paper as submitted